# Theoretical–Computational Modeling of CD Spectra of Aqueous Monosaccharides by Means of Molecular Dynamics Simulations and Perturbed Matrix Method

**DOI:** 10.3390/molecules28083591

**Published:** 2023-04-20

**Authors:** Massimiliano Aschi, Laura Palombi, Andrea Amadei

**Affiliations:** 1Dipartimento di Scienze Fisiche e Chimiche, Università dell’Aquila via Vetoio (Coppito 2), 67010 l’Aquila, Italy; 2Dipartimento di Scienze e Tecnologie Chimiche, Università di Roma “Tor Vergata”, Via della Ricerca Scientifica 1, 00133 Roma, Italy

**Keywords:** computational spectroscopy, electronic circular dichroism, condensed phase systems, molecular dynamics, quantum chemical calculations, perturbed matrix method

## Abstract

The electronic circular dichroism (ECD) spectra of aqueous d-glucose and d-galactose were modeled using a theoretical–computational approach combining molecular dynamics (MD) simulations and perturbed matrix method (PMM) calculations, hereafter termed MD-PMM. The experimental spectra were reproduced with a satisfactory accuracy, confirming the good performances of MD-PMM in modeling different spectral features in complex atomic–molecular systems, as already reported in previous studies. The underlying strategy of the method was to perform a preliminary long timescale MD simulation of the chromophore followed by the extraction of the relevant conformations through essential dynamics analysis. On this (limited) number of relevant conformations, the ECD spectrum was calculated via the PMM approach. This study showed that MD-PMM was able to reproduce the essential features of the ECD spectrum (i.e., the position, the intensity, and the shape of the bands) of d-glucose and d-galactose while avoiding the rather computationally expensive aspects, which were demonstrated to be important for the final outcome, such as (i) the use of a large number of chromophore conformations; (ii) the inclusion of quantum vibronic coupling; and (iii) the inclusion of explicit solvent molecules interacting with the chromophore atoms within the chromophore itself (e.g., via hydrogen bonds).

## 1. Introduction

The theoretical modeling of the electronic circular dichroism (ECD) of molecular systems in a solution has been at the focus of interest in recent years [1,2,3,4,5,6,7], not only for fundamental but also for practical reasons. Furthermore, the possibility of modeling ECD spectra would be essential for determining the absolute configuration of chiral compounds and the conformational transitions in polypeptides [8,9]. A valid computational protocol could be, then, of great importance for scientists working in different fields, ranging from organic chemistry to biochemistry and medicinal chemistry. The reliability of the calculations of the ECD spectra of molecules in a solution is not a trivial task, as it depends upon key aspects which, in a rough hierarchical scale, could be summarized as follows:First of all, it is essential to use accurate electronic structure theory calculations for determining the objects of interest, which, in this case, are the excitation energies and the associated electric and magnetic transition moments. Therefore, time dependent density functional theory [10] using exchange-correlated functionals, such as B3LYP [11], or range-separated functionals, such as CAM-B3LYP [12], or other related approaches [13,14,15] revealed a suitable and computationally affordable tool, though alternative and possibly more accurate strategies have emerged [16,17].Secondly, it is well known that the morphological features of the ECD signal is extremely sensitive to chromophore conformational transitions [1] and possibly to quantum vibronic coupling [18,19]. For this reason, particularly when non-rigid species are concerned, modeling of this kind of spectroscopy requires an accurate and exhaustive conformational analysis of the chromophore, prior to the electronic structure calculations (see previous item) by means of molecular dynamics (MD) simulations.Finally, and related to the previous item, the presence of the explicit solvent in the model is of primary importance because of its influence on the conformational repertoire of the chromophore and, most importantly, because of its possible direct perturbing effects on the chromophore’s electronic properties underlying the ECD signal [1,3,20].

Several investigators, including ourselves, have proposed theoretical–computational approaches for this purpose in recent years, using different schemes for extending the ECD modeling to complex molecular systems, such as: (i) the direct calculation on a number of structures efficiently extracted from MD simulations [21,22,23]; (ii) exciton coupling [24]; (iii) evolutionary algorithms [25]; and (iv) MD simulations followed by the perturbed matrix method (PMM), hereafter MD-PMM [26,27]. In the spirit of the method reported in a previous work [28] and, in particular, following the MD-PMM, as described in a recent paper [27], we herein present a study aimed at modeling the ECD spectrum of molecular systems in a solution using an approach that integrates MD simulations, essential dynamics (ED) analysis [29,30], and PMM calculations [31,32,33] for an exhaustive conformational analysis of the chromophore and also including the effect of the explicit solvent molecules in modeling the spectroscopic objects.

For this purpose, we have selected a particularly challenging and well-investigated topic represented by the reconstruction and interpretation of the ECD spectrum of two monosaccharides in an aqueous solution, namely d-glucose and d-galactose (Figure 1).

The ECD spectrum of d-glucose, i.e., the spectrum of alpha-D-glucopyranose (hereafter, Alpha-Glu) and beta-D-glucopyranose (hereafter Beta-Glu), has already been the focus of a large number of studies [34,35,36,37,38], and it has been recently reappraised by a joint experimental and computational study based on the calculation of the spectral signal using 40 structures extracted from an MD simulation, with the continuum dielectric mimicking the solvent effect [39]. Such a study, which successfully reproduced the experimental data, had in fact clearly demonstrated the very strong dependency of the ECD signal on d-glucose stereochemical features, in particular: (i) the alpha and beta configurations; (ii) the gauche (G) and trans (T) conformations generated by the torsion motion involving the hydroxymethyl group and pyranoside oxygen (i.e., the O6-C-C-O5 dihedral angle in Figure 1), producing three different states termed as GG, GT, and TG; and finally, (iii) the different rotamers associated with the torsions of all the hydroxyl groups. In principle, a large number of chromophore and solvent configurations were then conceivably necessary to obtain a proper reconstruction of the ECD signal. In this study, we showed that it is possible to capture the essential physical features of the ECD spectrum (i.e., both the position, the intensity, and the shape) by including via PMM the dynamical solvent perturbation and using, by means of Essential Dynamics analysis, a reduced, albeit statistically significant, number of chromophore reference structures.

## 2. Results

In this section, we focused our attention on the results concerning the modeling of the ECD spectrum, and therefore, we did not discuss the MD details used as they were not strictly related to the focus of this study. It would be sufficient for us to state that the conformational analysis, conducted on our simulations for d-glucose, produced similar results, as compared to those reported in the literature [39] on this widely investigated system [40,41,42,43,44].The results of the present study are entirely based on a theoretical framework, termed as MD-PMM, which has been extensively reported in our previous works [26,27,31,32,33,45] and here briefly reviewed. The method requires a preliminary MD simulation of the system of interest, e.g., a chromophore surrounded by the solvent molecules. If the chromophore is small enough [31], it could be directly identified as the quantum center (QC), i.e., the sub-portion of the simulated system whose electronic and general quantum properties is of interest, and its perturbations could be calculated along the entire MD trajectory. It is important to emphasize two key aspects of the method: (i) as compared to the typical quantum mechanic/molecular mechanic (QM/MM) methods [46], the QC and the rest of the system are simulated using the same Hamiltonian (typically. empirical force fields); (ii) the above definition of the QC, i.e., the simplest, in some cases, might be not suitable [47], and in these cases, its definition should be addressed critically and according to the problem at hand. More explicitly, the computational strategy of PMM is based on the a posteriori evaluation at each frame of the MD trajectory of the electronic and magnetic properties of the QC, along with the rest of the system, i.e., the solvent, acting as an electrostatic perturbation.

The electronic Hamiltonian operator $\hat{H}$ of the QC embedded in the perturbing environment can be expressed by the following:(1)$$\hat{H}={\hat{H}}^{0}+\hat{V}$$
where ${\hat{H}}^{0}$ is the QC unperturbed electronic Hamiltonian (i.e., as obtained and considering the isolated QC) and $\hat{V}$ is the perturbation operator. Within the PMM scheme, the perturbing electric field as provided by the environment (solvent) atomic charges is used to obtain the perturbation operator, $\hat{V}$, via a multi-polar expansion focused on the QC center of mass, $r_{0}$, as follows:(2)$$\hat{V}≅\sum_{n}\left[V\left(r_{0}\right)-E\left(r_{0}\right)\cdot \left(r_{n}-r_{0}\right)+\ldots \right]q_{n}$$
where *n* runs over all QC particles (i.e., nuclei and electrons), $q_{n}$ is the charge of the *n*-th particle, $r_{n}$ is the corresponding coordinates, V is the electrostatic potential exerted by the perturbing environment, and **E** is the perturbing electric field (E=−∂V/∂r). At each frame of the MD simulation, the perturbed electronic Hamiltonian matrix is then constructed and diagonalized, providing a continuous trajectory of perturbed eigenvalues and eigenvectors $c_{i}$ to be used for evaluating the QC instantaneous perturbed quantum object of interest: e.g., the electronic excitation energies and the electric ($\mu_{i,j}$) and magnetic ($m_{i,j}$) transition dipole moments for the $\Phi_{i}\to \Phi_{j}$ perturbed electronic eigenstate transition, with their *p* component given by the following:(3)$$\begin{array}{c} {\left[{\tilde{\mu }}_{p}\right]}_{i,j}=c_{i}^{*T}{\tilde{\mu }}_{p}^{0}c_{j} \end{array}$$(4)$$\begin{array}{c} {}[{\tilde{\mu }}_{p}^{0}{]}_{l,l^{\prime }}=\;\langle \Phi_{l}^{0}|{\hat{\mu }}_{p}|\Phi_{l^{\prime }}^{0}\rangle \end{array}$$(5)$$\begin{array}{c} {}[{\tilde{m}}_{p}{]}_{i,j}=c_{i}^{*T}{\tilde{m}}_{p}^{0}c_{j} \end{array}$$(6)$$\begin{array}{c} {}[{\tilde{m}}_{p}^{0}{]}_{l,l^{\prime }}=\;\langle \Phi_{l}^{0}|{\hat{m}}_{p}|\Phi_{l^{\prime }}^{0}\rangle \end{array}$$
where $c_{i}$ and $c_{j}$ are the unit vectors representing the $\Phi_{i}$ and $\Phi_{j}$ perturbed electronic eigenstates, respectively, within the unperturbed eigenstate basis set ($\Phi_{l}^{0}$), and ${\tilde{\mu }}_{p}$ and ${\tilde{\mu }}_{p}^{0}$ and ${\tilde{m}}_{p}$ and ${\tilde{m}}_{p}^{0}$ are the *p* component electric and magnetic dipole moment matrices expressing the corresponding electric (${\hat{\mu }}_{p}$) and magnetic (${\hat{m}}_{p}$) dipole operators within the perturbed and unperturbed basis set, respectively.

The CD signal could be obtained by evaluating the rotational strength $R_{k}$ from each ground (0) to *k* excited state transition as defined by:(7)$$R_{k}=Im\left\{\mu_{0,k}\cdot m_{0,k}\right\}$$
where Im{} indicates the imaginary part of the expression between the brackets.

Therefore, the CD spectral curve as a function of the frequency (ν) could be expressed via [27,45]: (8)$$\begin{array}{ccc} \Delta \varepsilon (\nu ) & = & \sum_{k}\Delta \varepsilon_{0,k}\left(\nu \right) \end{array}$$(9)$$\begin{array}{ccc} \Delta \varepsilon_{0,k}\left(\nu \right) & = & \sum_{\nu_{ref}}\frac{{\left(R_{k}\right)}_{\nu_{ref}}\;\nu }{2.296\times 10^{-39}}\;\frac{n(\nu_{ref})}{N}\;\frac{e^{-{\left(\nu -\nu_{ref}\right)}^{2}/\left(2\sigma^{2}\right)}}{\sqrt{2\pi \sigma^{2}}} \end{array}$$
where Δε is the entire molar circular dichroism spectrum, defined as the difference between the molar extinction coefficients of the left and right circularly polarized light; $\Delta \varepsilon_{0,k}$ is the molar circular dichroism spectrum for the 0→k electronic state transition; the summation in Equation (9) runs over the frequency bins used to construct the spectrum and identified by the reference frequencies $\nu_{ref}$ (i.e., the bin middle values); *N* is the total number of MD frames (necessarily a large number to obtain a reliable spectral signal); $n(\nu_{ref})$ is the number of MD frames with (vertical) electronic excitation frequency ν within the bin centered at $\nu_{ref}$; ${\left(R_{k}\right)}_{\nu_{ref}}$ is the (perturbed) mean rotational strength as obtained by averaging over the bin MD frames, as expressed in cgs units, and $2.296\times 10^{-39}$ is the conversion factor needed to obtain Δε in cgs units. Finally, the Gaussian distributions centered at each $\nu_{ref}$ provided broadening due to the classical vibrations disregarded by the PMM calculations [45] with the value of σ, as obtained by the procedure used in a previous paper [48]. We should note that this latter step was only required for smoothing the final result, whose essential features, e.g., the shape and intensity of the spectral signal, is automatically reproduced in the MD-PMM. The above theoretical framework was transformed into a computational strategy as schematically depicted in Figure 2, and briefly outlined as follows.

The investigation was initiated by simulating the monosaccharides (i.e., the chromophore) in the alpha and beta configurations separately, in a box filled with water molecules. These simulations (four in total) are hereafter termed as MD-free.The MD-free was then analyzed through the ED analysis (see below), which allowed us to evaluate the chromophore free-energy conformational landscape, i.e., the location of an M number of chromophore free-energy basins. Subsequently, it was possible to extract a number $N_{j},j=1,M$ of chromophore reference configurations (hereafter, RC) that were representative of the conformations to be used for each of the M free-energy basins, for a total of $N_{tot}$ reference structures. Note that a single RC might be not sufficient for describing a single conformational basin. Each of the $N_{tot}$ RC, corresponding to each conformation characterized by the relative free energy and probability provided by the MD-free simulation, was then independently simulated with a constrained MD simulation (i.e., the chromophore is kept frozen), hereafter termed MD-constr, in order to increase the solvent conformational sampling included in the modeling of the spectral features (see the previous subsection). Note the all the $N_{tot}$ MD-constr, although carried out exactly under the same MD-free conditions did not require for a proper sampling of the same simulation length because only the fast solvent relaxation had to be achieved.Each of the $N_{tot}$ RC was used for the quantum chemical calculations during the gas phase. In particular, the RC was first optimized by retaining the semi-classical internal degrees-of-freedom (proper dihedrals). Subsequently, the constrained-optimized structure was used for determining the unperturbed properties necessary for the PMM calculations (see the Theory Section) with the QC corresponding to the entire chromophore. Note that in order to test the performance of the method even when using the most simplified QC definition, we did not include any water molecules within the QC [28], though their presence was possible.The unperturbed properties of each RC were used for the PMM calculations of the corresponding MD-constr, providing the perturbed rotational strength, and the conformation CD spectrum (see Equation (Equation 9) in the Theory Section).The total spectra of the chromophore for the alpha and beta configurations were then evaluated by summing all the $N_{tot}$ spectra, each weighted by the corresponding conformation probability, as obtained by the MD-free simulation.Finally, the overall monosaccharide spectrum was obtained by summing the alpha and beta spectra weighted by the corresponding probability taken from the literature (see below) [49,50].

### 2.1. d-Glucose ECD Spectrum

As shown in Figure 3, the spectrum of the eigenvalues of the all-atom covariance matrix as obtained from MD-free simulations for Alpha-Glu and Beta-Glu clearly indicated that the entire chromophore fluctuation is concentrated in the first two eigenvectors in both of the configurations. The analysis of the atomic composition of these two eigenvectors, reported in the Appendix A, revealed that, as expected, the conformational fluctuations is driven by the rotations of the hydroxyl and hydroxymethyl groups. This finding, as mentioned in the previous section, allowed us to project the MD-free chromophore coordinates onto a two-dimensional plane, providing a straightforward view of the corresponding free-energy landscapes (see Equation (Equation 10)), reported in Figure 4.

For the sake of clarity, we have also reported, in the same figure, the mean values of the O6-C-C-O5 dihedral angle for the hydroxymethyl rotation (Figure 1) as a function of the same two-dimensional (essential) plane position, as obtained by the MD-free (note that according to our definition, GT, GG, and TG correspond to the dihedral angle approximately equal to 180${}^{\circ }$, 60${}^{\circ }$/240${}^{\circ }$, and 0${}^{\circ }$, respectively.) Results reported in the figure show that, in good agreement with experimental estimations and even with recent simulations [39] on the same system, the probability of the TG state was negligible and that Alpha-GT:Alpha-GG = 42:58 and Beta-GT:Beta-GG = 40:60.

Eleven RCs were extracted within the lowest free-energy regions of the three conformational basins, as evidenced by the ED analysis (see Figure 4), both for Alpha-Glu and Beta-Glu. A schematic view of the RCs of Alpha-Glu and Beta-Glu are shown in Figure 5 and Figure 6.

Additional details, including the locations of the RCs in the essential plane and their Cartesian coordinates, are reported in the Appendix A. In Table 1, we detailed the probabilities of the conformations corresponding to the reference structures, as obtained from the ED analysis.

In order to carry out a preliminary test of our computational setup (both the MD simulation and unperturbed basis set for PMM), we evaluated the absorption UV spectrum of aqueous d-glucose. This was carried out following the same PMM-based strategy, which had been widely described in our previous studies (for example, reference [45]) and not reported here in the interest of brevity. For this purpose, we calculated the absorption UV spectrum for each of the RC conformations via the corresponding MD-constr. These spectra were then weighted by the probabilities of the RC conformations (see Table 1) and summed to obtain the complete absorption spectrum of the alpha and beta configuration. Finally, the two obtained spectra were weighted by the alpha and beta (experimental) probabilities and summed to obtain the results reported in Figure 7. It is well known that in a water solution, it is only possible to highlight a sharp red tail between 200 and 220 nm. Our calculations (see Figure 7) efficiently reproduced this feature, which provided a further suggestion of the consistency of the present computational setup, though not decisively.

In order to obtain the complete ECD spectrum of d-glucose, we first calculated, by means of each MD-constr, the corresponding ECD spectrum (see the Theory Section and computational details), as reported in Figure 8 for all the RCs. As shown in the figure, it was evident that for both the alpha and beta configurations that each conformation had a specific spectrum, thus showing the high variability of the ECD spectral signal due to minimal structural variations.

Note that the wavelength region-of-interest, i.e., the spectral domain actually sampled in the experiments, fell between 160 and 220 nm. For this reason, in the complete calculated ECD spectra, we disregarded the wavelengths below 160 nm.

From the weighted sum (see Table 1) of the RC conformation spectra reported in Figure 8, we evaluated the spectrum of the Alpha-Glu and Beta-Glu, and finally, by using the experimental Alpha-Glu:Beta-Glu = 32:68 ratio [35], we obtained a complete d-glucose ECD spectrum, hereafter indicated as the overall spectrum. The calculated Alpha-Glu and Beta-Glu spectra, as well as the overall spectrum, are shown in Figure 9, where they are compared to the corresponding experimental spectra [35].

We should emphasize the contribution of each perturbed electronic transition to the ECD spectrum. For this purpose, in Figure 10 and Figure 11, we reported all the RC conformations of the alpha and beta d-glucose and the weight of each ground–excited (perturbed) state transition, as evaluated by the maximum intensity of the corresponding spectral signals (Equations (8) and (9)). For clarity, we considered 3 different spectral ranges, i.e., 185–220 nm (the red tail of the spectrum), 165–175 nm (the wavelength region of the spectral maximum), and the intermediate range, 175–185 nm. We observed that the ECD spectra could be considered the result of a relatively high number of transitions (at least nine). Moreover if we limited our attention to the most interesting region, i.e., 166–175 nm, we could identify that at least 7 states (from 3rd to 9th) that provided a significant contribution to the spectra of all the RCs considered.

Based on the comparison of the calculated spectra with the available experimental data [35,39], we could draw the following conclusions.

The maximum of the global ECD spectrum was found at 174 nm, i.e., 6 nm red-shift with respect to the experimental value. This result was probably due to the intrinsic limitations of the CAM-B3LYP functional that had provided the same shift in the study of Matsuo and Gekko [39]. Not surprisingly, for cases such as the present one in which the vibronic effect had not been taken into account and where we had used the vertical transition approximation [45], the absolute intensity was overestimated, though the correct order-of-magnitude of the experimental spectral intensity [35] was reproduced.The relative heights of Alpha-Glu and Beta-Glu (1:0.55) ECD spectra, as obtained by our model, were in good agreement with the experiments (1:0.54 [39] and 1:0.60 [35]) and the overall spectral asymmetric shape [35,39] was properly reproduced. However, the calculated Beta-Glu spectrum and, consequently, the calculated overall spectrum appeared as characterized by a red-tail that faded at 15 nm (see Figure 9) beyond the experimental value. The origin of such a slight discrepancy could have been be twofold. On one hand, by using a limited number of RC conformations, as obtained by the MD sampling, our model could have suffered from some the inadequacies in the force field, resulting in an inaccurate sampling of the conformational regions. On the other hand, as already remarked in some of our recent studies [47], the purely classical QC-solvent interactions we used to model the QC perturbation could have resulted in incorrect hydrogen-bonding, leading to the overestimation of the first solvent shell mobility and, thus, to overestimated electric field fluctuations.

### 2.2. d-Galactose ECD Spectrum

The ECD spectrum of the aqueous d-galactose, i.e., the spectrum of alpha-d-galactopyranose (hereafter, Alpha-Gal) and beta-d-galactopyranose (hereafter, Beta-Gal), has not received any attention, as compared to d-glucose. Nevertheless, we applied the same strategy, as previously outlined, for modeling the ECD spectrum also of this monosaccharide, as well. Moreover, as an additional test of the efficiency and the accuracy of the method, we decided to further reduce the number of RCs used for defining the relevant conformations. Note that for the sake of brevity in the last subsection, we limited our attention to the main results and did not present and discuss all the data obtained (additional information is reported in the Appendix A). The ED analysis revealed (see Appendix A) a spectrum of the covariance matrix eigenvalues, as well as the eigenvector composition, that could be superimposed over that for d-glucose. For this reason, a conformational analysis was carried out in the space of the first two eigenvectors. As expected, as shown in the upper panels of Figure 12, we observed that the free-energy landscape of galactose was characterized by the presence of three conformational basins corresponding to the GG and GT rotamers (see the lower panels of the Figure 12). In this case, we decided to extract seven reference structures for the alpha configuration and seven reference structures for the beta configuration. Schematic views of these structures are reported in Figure 13 and Figure 14, and the probabilities (as obtained from Figure 12) are described in Table 2.

The obtained ECD spectrum, as compared to the experimental results, is shown in Figure 15.

Furthermore, in this case we can draw the following considerations

The morphological features of the spectrum in the region-of-interest appeared to be sufficiently well reproduced. In addition, both the signs of the Alpha-Gal and Beta-Gal signals, as well as their shapes, were in satisfactory agreement with the experimental data. However, as compared to Glu, in the case of Alpha-Gal, a slight but significant deviation from the experimental spectrum was observed in two spectral regions, i.e., below 170 nm and between 185 and 190 nm. Furthermore, in these regions, our model predicts two negative peaks that were absent in the experimental spectrum, probably due to a slight inadequacy in the utilized conformational repertoire. It was interesting to observe that the positive sign of Alpha-Gal was determined by the GT rotamer (see Appendix A for additional details).The Alpha-Gal and Beta-Gal minimum and maximum spectra, on the other hand, as compared to the experimental values, appeared slightly red-shifted (+2.5 nm) and blue-shifted (−5.0 nm) for Alpha-Gal and Beta-Gal, respectively.The calculated overall spectrum (obtained using the experimental Alpha:Beta population ratio of 38:62) showed the main features of the experimental results, although with enhanced intensity variations within the wavelength range-of-interest, probably due to the slightly overestimated absolute intensity, as briefly discussed for d-glucose.Nevertheless, even using a drastically reduced number of RC conformations, our model was able to capture the essential spectral features of D-Galactose.

## 3. Materials and Methods

All the MD simulations were carried out in the NVT ensemble with a time-step of 2.0 fs with the program Gromacs [51,52] version 5.1.2. The chromophore was inserted at the center of a cubic box (9.24 nm${}^{3}$ volume) filled with 951 water molecules [53]. The density of the system (i.e., the box size) was adjusted to reproduce the average pressure obtained from a simulation of an identical box containing only the same number of water molecules at the experimental density of 298 K and 1.0 bar. The monosaccharides were described using the all-atom Gromos force field [54] (G53a6 version) with the parameters obtained from the ATB repository [55]. The quality of the force field was ascertained a posteriori, considering the distribution of the rotamers (see below) in the MD-free, which had been widely investigated experimentally and computationally [56]. The temperature was kept constant using the Parrinello thermostat [57], and the bond lengths were constrained using the LINCS algorithm [58] and long-range electrostatic interactions were computed by the particle mesh Ewald method [59] with 34 wave vectors in each dimension and a 4th-order cubic interpolation. Finally, a cut-off of 1.1 nm was used. The simulations (for Alpha-Glu, Beta-Glu, Alpha-Gal, and Beta-Gal) were extended up to 100.0 ns. The MD-constr simulations were extended up to 20.0 ns using the same force fields and computational protocol of MD-free, except for freezing the solute molecule. The ED analysis was performed as follows. The chromophore structure was first (mass-weighted) fitted to a reference structure in order to remove the roto-translational motions from the analysis. Subsequently, the all-atom covariance matrix was first constructed and then diagonalized to obtain a set of eigenvalues corresponding to the mean-squared fluctuations of the generalized coordinates, as defined by the eigenvectors.The eigenvectors corresponding to the largest eigenvalues, typically termed as essential eigenvectors, defined the subspace (essential space) in which it was possible to describe the chromophore conformational transitions. For this purpose, the chromophore Cartesian coordinates, as obtained by the MD-free time frames were projected onto the essential eigenvectors. This procedure allowed us to identify the regions of the essential space (i.e., the conformations) characterized by a non-zero probability *P*. Consequently, we could estimate the Helmholtz free-energy difference ($\Delta A_{i,j}$) between the *i* and *j* conformations through the standard equation as follows:(10)$$\Delta A_{i,j}=-k_{B}TlnP_{j}/P_{i}$$

All the unperturbed quantum–chemical calculations were performed in the framework of the density functional theory using the CAM-B3LYP [12] functional, which, as previously remarked, had revealed to be a rather suitable tool for this purpose. The 6-31+G* basis set was used. All the RCs were first optimized with internal constraints (see above) and then, using the obtained geometries, a set of 16 unperturbed electronic excited (vertical) states were evaluated using the time-dependent density functional theory. For each of the extracted RCs, the set of 17 electronic states provided the unperturbed basis set for the PMM calculations. All the calculations were performed with the Gaussian 16 package [60]. Additional details about the optimized geometries and corresponding properties for PMM calculations are reported in the Appendix A.

## 4. Discussion and Concluding Remarks

The modeling of the ECD spectra was a rather challenging task for theoretical–computational chemistry. This was primarily due to the high sensitivity of the ECD signal to the chromophore and the solvent configurations, as this imposed the calculation of the spectral signal through quantum–chemical methods on a prohibitively high number of solvent–solute conformations. For this reason, several strategies have been proposed in recent years for obtaining physically consistent results by avoiding such a brute-force approach. Within this context, we applied a theoretical–computational approach for quantitatively modeling the ECD spectrum of d-glucose and d-galactose in water. Based on a careful preliminary conformational search performed through MD simulations and an essential dynamics analysis followed by the explicit ECD modeling via PMM calculations, our model was able to reconstruct the spectral signal based on the configured fluctuations of the chromophore–solvent system. According to the results obtained for both monosaccharides and confirming the results obtained by recent studies [27,28], MD-PMM could be considered a suitable tool and an alternative to other theoretical–computational methodologies, for the modeling of the ECD spectra of chromophores in a solution. In particular, our primary aim was to show that it was possible to capture the essential quantitative features of the ECD spectrum of a flexible chromophore, not only using a reduced number of conformational states but also by adopting the simplest definition of the quantum center, i.e., without including the presence of any solvent molecules. The satisfactory results obtained for both systems clearly indicated the advantages of this method, which could be briefly summarized in a few points: First, the high internal physical consistency of the model allowed us to include, rigorously and explicitly, the semi-classical effects of the solvent molecules and the actual weight of the chromophore conformational states. Second, our approach had a relatively low computational cost, thought it is worth mentioning that application of the MD-PMM approach required special attention, in particular, concerning two points: (i) according to the definition of the QC employed, we had to critically assess the consistency and accuracy of the PMM level of theory adopted, e.g., the use of the dipole approximation for modeling the perturbed Hamiltonian operator was not suitable for large QCs where the homogeneous perturbing field approximation was inaccurate (see, for example, reference [28]), and thus, higher levels of PMM theory had to be employed [33]. (ii) the consistency and accuracy of the final results were strongly connected not only to the level of the electronic structure calculations but also, in particular, to the quality of the adopted atomistic empirical force field of the MD simulations.

## Figures and Tables

**Figure 1 molecules-28-03591-f001:**
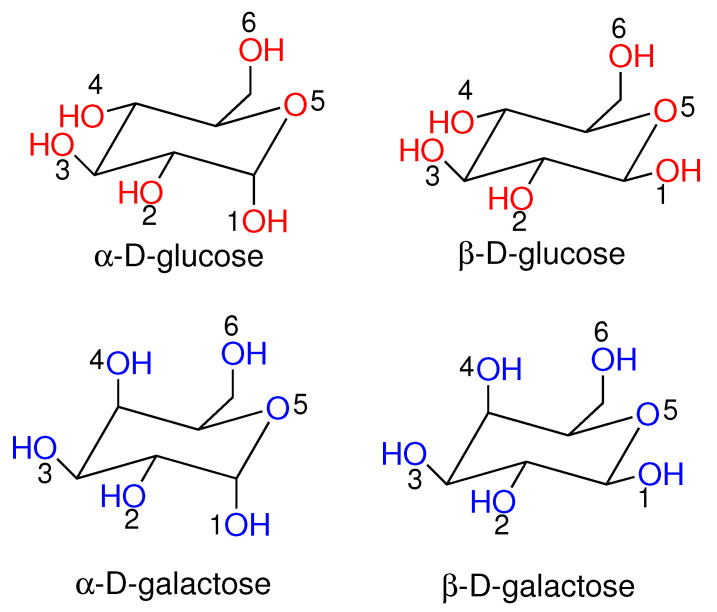
Schematic view of D-gluglucopyranose (alpha and beta) and D-galactopyranose (alpha and beta).

**Figure 2 molecules-28-03591-f002:**
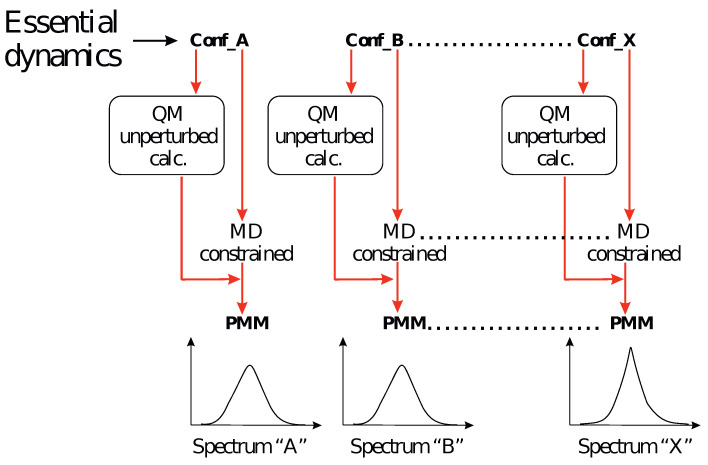
Schematic view of the entire computational procedure (following the MD simulation) utilized in the present work. Based on the essential dynamics analysis, a number of representative conformations (Conf) were extracted and used for (i) unperturbed quantum chemical calculations and (ii) constrained MD simulations. PMM was then applied for each of the extracted Conf for modeling the related spectra. The final spectrum was obtained by statistically weighting the spectra.

**Figure 3 molecules-28-03591-f003:**
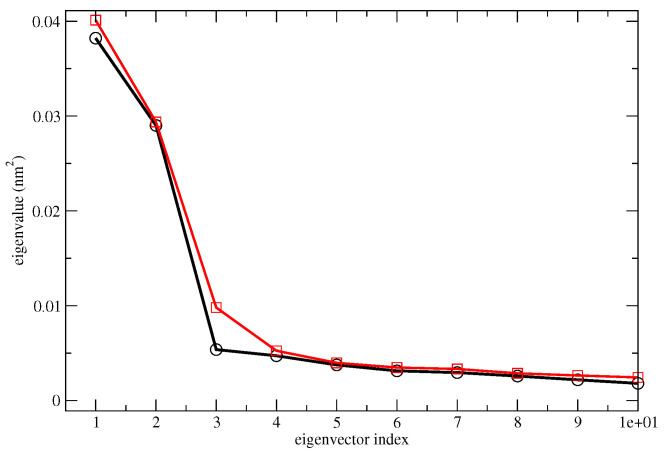
Spectrum of the eigenvalues of the all-atoms covariance matrix for the Alpha-Glu (black) and Beta-Glu (red).

**Figure 4 molecules-28-03591-f004:**
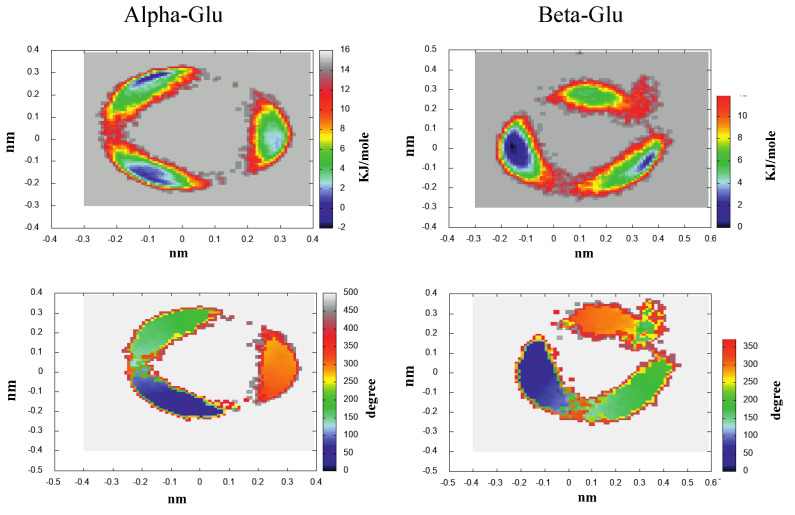
300 K Helmholtz free-energy variation in the essential plane for Alpha-Glu (**left-upper** panel) and Beta-Glu (**right-upper** panel). The O6-C-C-O5 torsion angle (mean) value as a function of the essential plane position for Alpha-Glu (**left-lower** panel) and Beta-Glu (**right-lower** panel).

**Figure 5 molecules-28-03591-f005:**
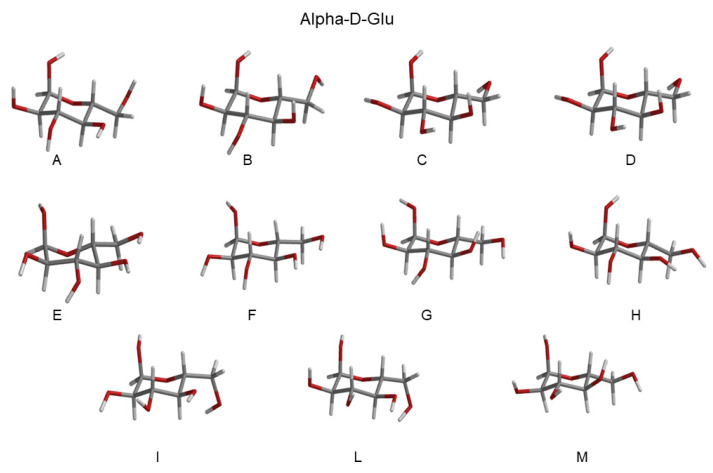
Schematic view of the Alpha-Glu RCs extracted from the basins, as reported in Figure 4. Additional details can be found in the Appendix A.

**Figure 6 molecules-28-03591-f006:**
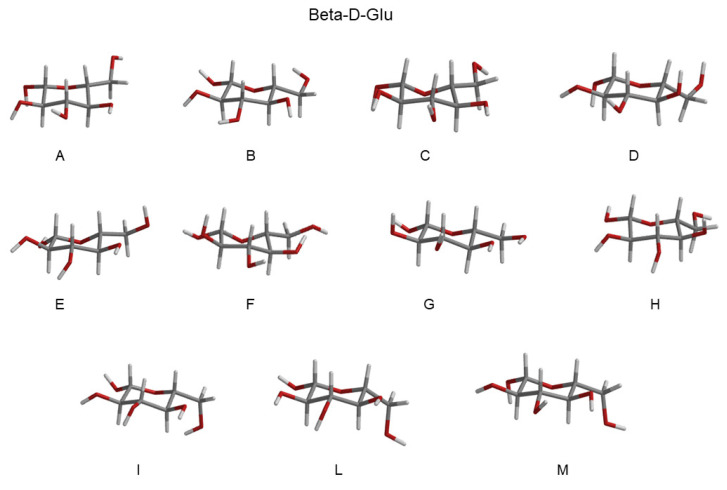
Schematic view of the Beta-Glu RCs extracted from the basins reported in Figure 4. Additional details can be found in the Appendix A.

**Figure 7 molecules-28-03591-f007:**
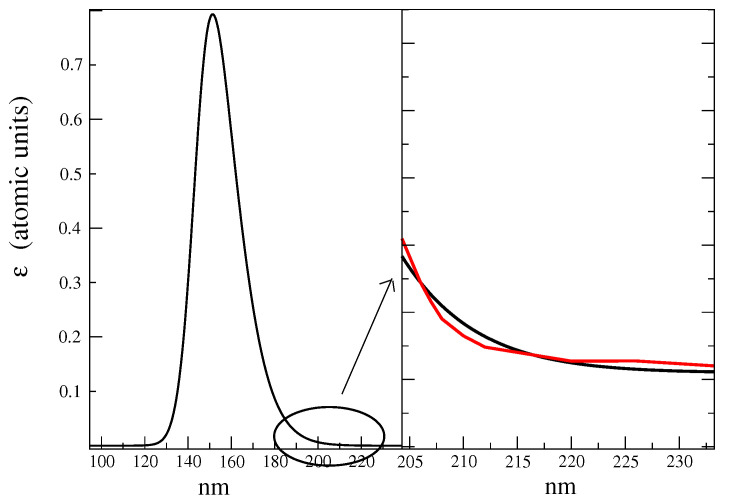
Left panel: Calculated UV absorption spectrum of d-glucose in a water solution. Right panel: Detail of the highlighted red-tail region of the calculated (black) and experimental (red) UV absorption spectrum of d-glucose in a water solution.

**Figure 8 molecules-28-03591-f008:**
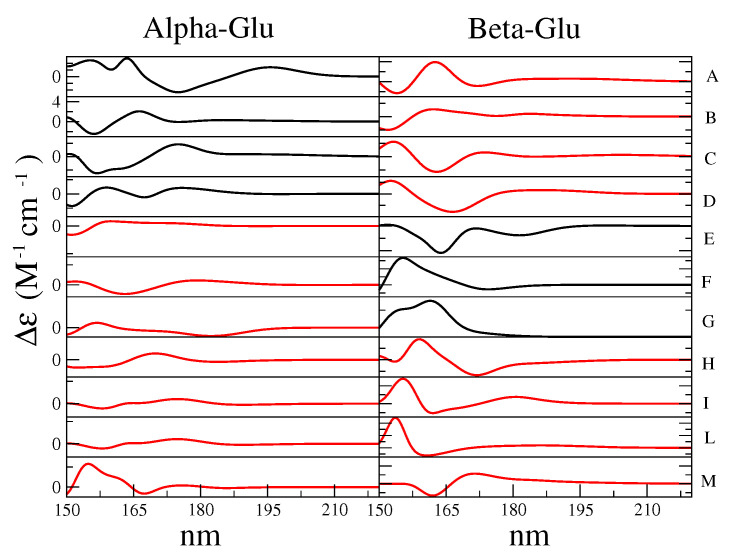
Calculated ECD spectra of the RC conformations reported in Table 1. For the sake of clarity, we have omitted the values on the y-axis. The names of the species, reported on the right side, are the same as in Table 1. In red, we have indicated the GG rotamers, and in black, the GT rotamers.

**Figure 9 molecules-28-03591-f009:**
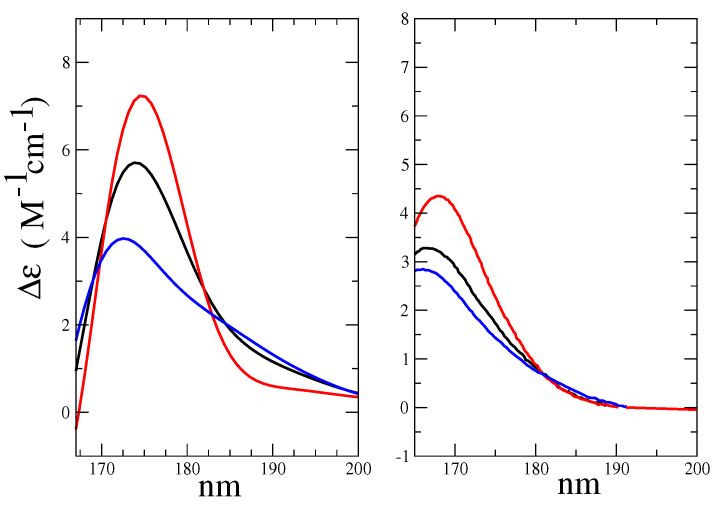
**Left side**: Calculated ECD spectra of Alpha-Glu (red), Beta-Glu (blue), and the overall spectrum (black), in a water solution. **Right side**: The experimental spectra [35] of Alpha-Glu (red), Beta-Glu (blue), and the total spectrum (black), in a water solution.

**Figure 10 molecules-28-03591-f010:**
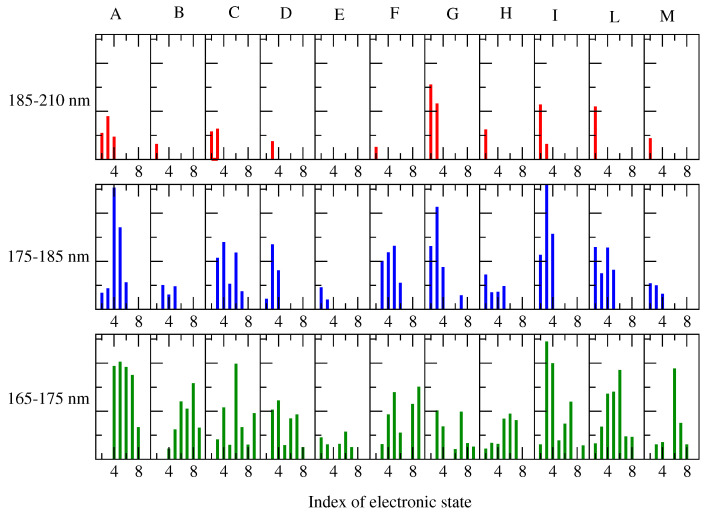
Weight of each of the electronic ground-excited (perturbed) state transitions, involved in the calculated ECD spectra of Alpha-Glu for each RC conformation (indicated in the upper side), as obtained by considering their perturbed maximum intensities. Note that for clarity, only the transitions showing a maximum intensity higher than 1.0 M ${}^{-1}$ cm ${}^{-1}$ were taken into account.

**Figure 11 molecules-28-03591-f011:**
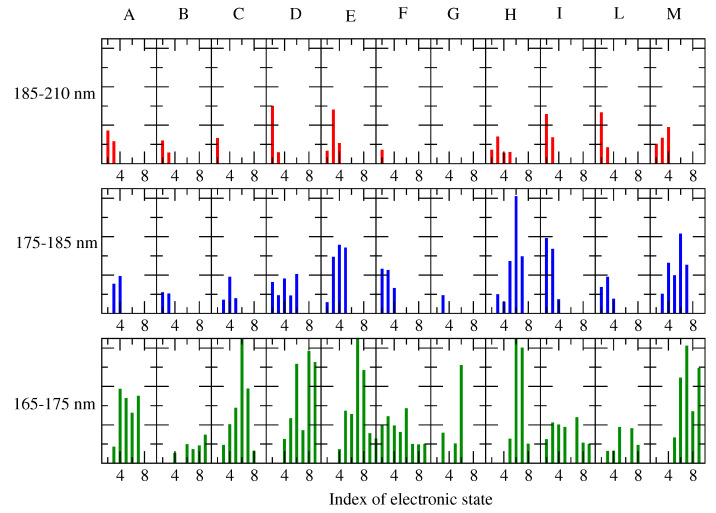
The electronic ground–excited (perturbed) state transitions involved in the calculated ECD spectra of Beta-Glu, as obtained for each RC conformation. Note that for clarity, only the transitions showing a maximum intensity higher than 1.0 M ${}^{-1}$ cm ${}^{-1}$ were taken into account.

**Figure 12 molecules-28-03591-f012:**
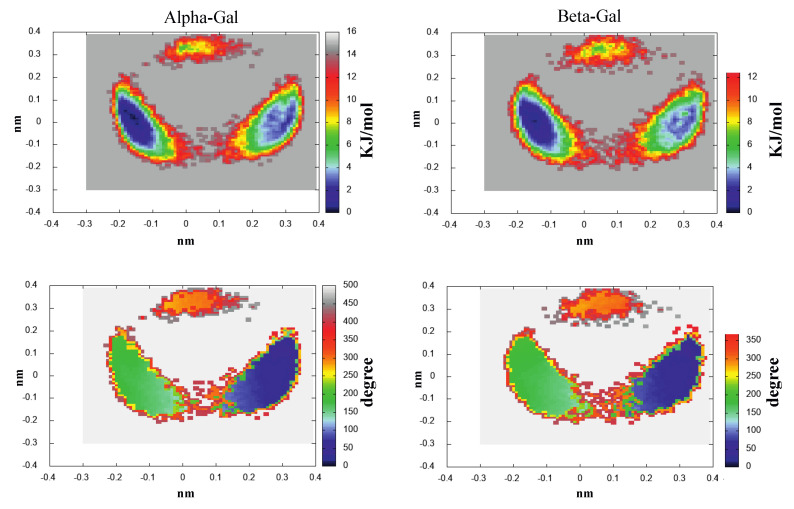
300 K Helmholtz free-energy variation in the essential plane for Alpha-Gal (**left-upper** panel) and Beta-Gal (**right-upper** panel). The O6-C-C-O5 torsion angle (mean) value as a function of the essential plane for Alpha-Gal (**left-lower** panel) and Beta-Gal (**right-lower** panel).

**Figure 13 molecules-28-03591-f013:**
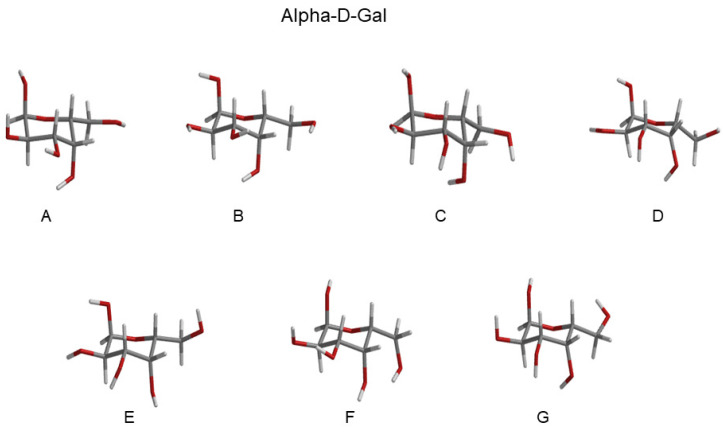
Schematic view of the Alpha-Gal RCs extracted from the basins, as reported in Figure 12. Additional details can be found in the Appendix A.

**Figure 14 molecules-28-03591-f014:**
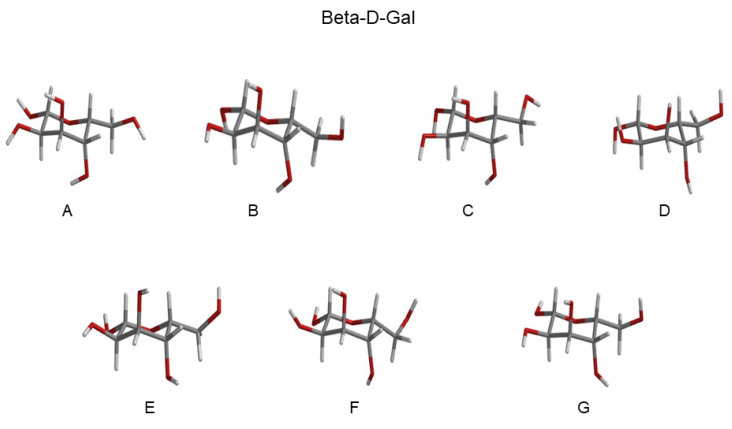
Schematic view of the Beta-Gal RCs extracted from the basins, as reported in Figure 12. Additional details can be found in the Appendix A.

**Figure 15 molecules-28-03591-f015:**
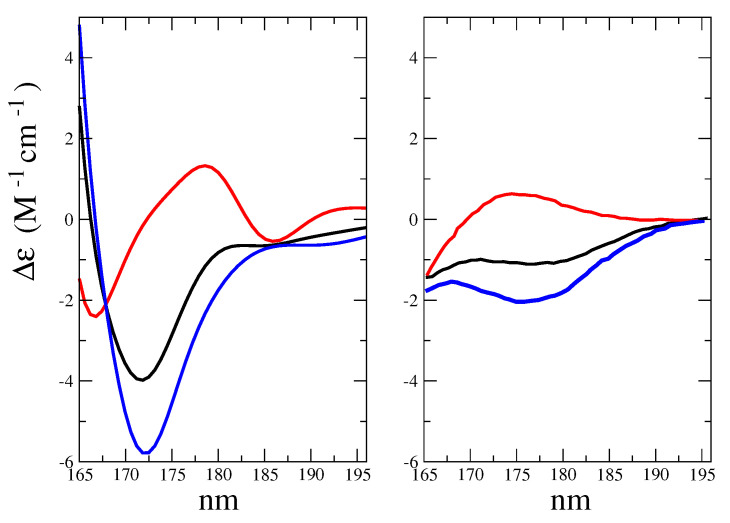
**Left side**: Calculated ECD spectra of Alpha-Gal (red), Beta-Gal (blue), and total (black), in a water solution. **Right side**: Reproduction of the experimental spectra from reference [35] of Alpha-Gal (red), Beta-Gal (blue), and total (black), in a water solution.

**Table 1 molecules-28-03591-t001:** Probability of the conformations (from A to M) corresponding to the RCs, as obtained by the MD-free for Alpha-Glu and Beta-Glu. The rotamer involved in each conformation is indicated in parentheses.

RC-Conf Alpha-Glu	Probability	RC-Conf Beta-Glu	Probability
A (GT)	0.10	A (GG)	0.10
B (GT)	0.10	B (GG)	0.10
C (GT)	0.09	C (GG)	0.11
D (GT)	0.13	D (GG)	0.10
E (GG)	0.10	E (GT)	0.13
F (GG)	0.09	F (GT)	0.14
G (GG)	0.09	G (GT)	0.13
H (GG)	0.08	H (GG)	0.07
I (GG)	0.10	I (GG)	0.04
L (GG)	0.10	L (GG)	0.04
M (GG)	0.11	M (GG)	0.04

**Table 2 molecules-28-03591-t002:** Probability of the conformations (from A to G) corresponding to the RCs, as obtained by the MD-free for Alpha-Gal and Beta-Gal. The rotamer involved in each conformation is indicated in parentheses.

RC Alpha-Gal (Basin)	Probability	RC Beta-Gal (Basin)	Probability
A (GT)	0.15	A (GT)	0.09
B (GT)	0.15	B (GT)	0.16
C (GT)	0.12	C (GT)	0.13
D (GT)	0.15	D (GT)	0.12
E (GG)	0.19	E (GG)	0.19
F (GG)	0.19	F (GG)	0.18
G (GG)	0.05	G (GG)	0.07

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
