# Peer review of "Theoretical–Computational Modeling of CD Spectra of Aqueous Monosaccharides by Means of Molecular Dynamics Simulations and Perturbed Matrix Method"

_molecules, 2023, doi:10.3390/molecules28083591_

Round 1
Reviewer 1 Report
This is a good paper, which demonstrates the ability of theoretical calculations to reproduce CD spectra, as illustrated with gluco- and galactopyranose. The perturbed matrix method with DFT was used to calculate the spectra of individual conformations, which were then combined into the ensemble-averaged spectrum of a respective system by means of MD simulations post-processed with principal component analysis. The agreement between theory and experiment is good and the respective theoretical and experimental epsilon(lambda) curves have the same topological features. The underlying methodology is sound and the calculations and analysis were performed adequately.
I have two minor suggestions:
(1) Compare the results obtained in this work with those by other researchers if any such calculations have been performed.
(2) Label the spectra in Figure 6 with the conformation labels of Table 1.
Reviewer 2 Report
Improving the theoretical calculation method of biomolecule CD spectra is very attractive topic because CD is significantly sensitive to the configurations of chromophores of small molecules such as monosaccharides. The CD calculations of monosaccharides have been discussed by a lot of papers, and in this paper, authors suggested new approach of the calculations. The theoretical techniques of new approach are based on the previous papers and it seems that the introduction of new trials such as PMM and RC are very adequate for the CD calculations of monosaccharides. There are some following requirements and questions for the publications.
1. The graphical pictures of initial structures and all RC structures of alpha- and beta- Glu and alpha- and beta-Gal should be described in the paper (or supplementary file). The visual information is also useful for promoting the understandings of unique configurations of each monosaccharide.
2. They estimated the populations of GG and GT rotamers in Glu and Gal from MD simulations. Are these estimated values close to those of experimental results? If not close, there are any limitations in your MD simulation? Further, can you compare with previous theoretical results on GG and GT populations? and describe any discussion on the differences and similarity? The comparisons with previous results would be necessary to understand the usefulness of the computational method of this paper.
3. In Figure 5 (left panel), the calculated UV spectrum showed zero below 140 nm. Is this correct?
4. To clearly show the differences and similarity between calculated and experimental spectra of Glu (or Gal), the vertical range in the left and right panels of Figure 7 should be same range. This should be conducted in the case of Gal (Figure 11). Further, the wavelength range of both figures should be extended to 220 or 210 nm (like Figure 6), (which means the range of wavelength should be from around 210 or 220 nm to around 165 nm. There is small negative peak CD in alpha-Gal around 185 nm. You can note the peak in the text and discuss on this unique peak.
